# Postoperative Geriatric Nutritional Risk Index as a Determinant of Tolerance to S-1 Adjuvant Chemotherapy After Curative Surgery for Pancreatic Ductal Adenocarcinoma: A Cohort Study with External Validation

**DOI:** 10.3390/cancers17091448

**Published:** 2025-04-26

**Authors:** Naotake Funamizu, Akimasa Sakamoto, Shozo Mori, Miku Iwata, Mikiya Shine, Chihiro Ito, Mio Uraoka, Yoshitomo Ueno, Kei Tamura, Yoshiaki Kamei, Yasutsugu Takada, Taku Aoki, Yuzo Umeda

**Affiliations:** 1Department of Hepato-Biliary Pancreatic and Breast Surgery, Ehime University Graduate School of Medicine, Shitsukawa 454, Toon-City 791-0295, Japan; sakamoto.akimasa.kw@ehime-u.ac.jp (A.S.); miku.nkgw@gmail.com (M.I.); takoyaking1018@gmail.com (M.S.); chippy.ito@gmail.com (C.I.); uraoka.mio.lr@ehime-u.ac.jp (M.U.); ueno.yoshitomo.xr@ehime-u.ac.jp (Y.U.); k-tamura@m.ehime-u.ac.jp (K.T.); kamei.yoshiaki.mz@ehime-u.ac.jp (Y.K.); takada-yasutsugu@yamatokoriyama.jcho.go.jp (Y.T.); umeda.yuzo.oe@ehime-u.ac.jp (Y.U.); 2Department of Hepato-Biliary Pancreatic Surgery, Dokkyo Medical University, Kitakobayashi 880, Mibu, Shimotsugagun 321-0293, Japan; shozomori@hotmail.co.jp (S.M.); aoki-2su@dokkyomed.ac.jp (T.A.)

**Keywords:** geriatric nutritional risk index, S-1, adjuvant chemotherapy, pancreatic ductal adenocarcinoma, validation study

## Abstract

Adjuvant chemotherapy (AC) with S-1 has produced promising results in patients with pancreatic ductal adenocarcinoma (PDAC). However, the completion rate of S-1 therapy remains low, limiting its full therapeutic potential. Furthermore, reliable markers for predicting S-1 treatment completion are lacking, prompting research into the geriatric nutritional risk index (GNRI) as a potential indicator of nutritional status. Our previous study found that a postoperative GNRI value of < 94.4 predicted S-1 AC noncompletion due to adverse events (AEs). This study aims to validate the link between the postoperative GNRI and S-1 therapy noncompletion because of AEs in an independent validation cohort from another institution.

## 1. Introduction

When diagnosed with pancreatic ductal adenocarcinoma (PDAC), only about 20% of patients are eligible for surgical intervention, which is the only curative option. Despite advances in diagnostic methods, chemotherapy, and surgical techniques, the 5-year survival rate remains at an average of 11.5% [1]. For those with resectable PDAC, surgical resection remains the cornerstone of treatment, and advances in adjuvant chemotherapy (AC) have significantly improved long-term survival rates [2]. Several clinical trials have shown that the combination of radical surgery and AC provides a considerably better prognosis than surgery alone [3,4]. In Japan, oral S-1 is now the standard AC for patients undergoing radical resection for PDAC, as recommended by national guidelines [5]. This recommendation is based on the JASPAC01 trial conducted by the Japan Adjuvant Study Group of Pancreatic Cancer, which found that S-1 therapy resulted in a significantly higher median and 5-year survival rates in postoperative patients than gemcitabine [6].

Despite AC’s proven efficacy in improving outcomes for patients with PDAC, the successful completion of therapy remains difficult due to postoperative complications (POCs) and adverse events (AEs) associated with the AC regimen [7]. According to institutional data and previous studies, S-1 therapy is completed at a rate of about 70% [6,8]. Consequently, risk factors for S-1 noncompletion must be identified and managed promptly to optimize treatment outcomes.

The Geriatric Nutritional Risk Index (GNRI) is now recognized as a useful tool for assessing nutritional status, predicting POCs, and determining their clinical implications [9,10,11,12]. This index, which is calculated using serum albumin levels and body weight relative to height, is both accessible and affordable, making it a viable option in various clinical scenarios. A previous study found that a postoperative GNRI below a certain threshold accurately predicted the risk of S-1 AC noncompletion due to AEs in patients with PDAC who underwent curative surgery. These findings demonstrated a strong link between postoperative nutritional status and the feasibility of completing S-1 therapy [13].

To validate these findings, the current study investigates whether the postoperative GNRI can serve as a predictor of tolerance to S-1 AC in patients who have undergone curative surgery for PDAC. The central research question is whether the GNRI is associated with the likelihood of completing S-1 therapy after surgery. We hypothesize that patients with a low postoperative GNRI are more likely to experience AEs, leading to an early discontinuation of S-1 AC treatment. By confirming this association in an external validation cohort, this study aims to enhance risk stratification and enable the early identification of patients at risk of treatment intolerance, ultimately contributing to improved strategies for managing AC in PDAC.

## 2. Materials and Methods

### 2.1. The Study Population

This study retrospectively examined an external cohort of 309 patients who underwent curative resection for PDAC at Dokkyo Medical University Hospital between January 2010 and March 2023. Notably, 129 patients were excluded from the analysis; 69 due to personal decisions to avoid S-1 AC, and 60 due to the use of alternative chemotherapies (Figure 1).

The final analysis included 180 patients, with no reported deaths within the first 90 days after surgery. Detailed patient data were gathered, including demographic information, perioperative laboratory findings, clinical records from the perioperative period, pathological diagnoses, and postoperative outcomes.

Importantly, the study met ethical standards and was approved by the institutional review board at Ehime University Hospital (approval number EUH2502008). The research followed the principles outlined in the Declaration of Helsinki (2013 revision). All participants, including those who were enrolled retrospectively or through legal representatives, provided explicit informed consent for the use of their medical data for research purposes.

### 2.2. Surgical Procedures and Perioperative Care at the Dokkyo Medical University Hospital

Pancreatic anastomosis during pancreatoduodenectomy was primarily performed using the end-to-side pancreatojejunostomy technique. During distal pancreatectomy, the pancreas was typically resected with a linear stapler. Closed suction drainage tubes were commonly used, with the number (two or three) determined by the surgeon’s preferences. POCs were classified using the Clavien–Dindo (CD) grading system, with CD grades ≥3 indicating major complications requiring significant intervention [14].

### 2.3. Adjuvant Chemotherapy Protocols and Postoperative Monitoring

At the Dokkyo Medical University Hospital, AC was started shortly after being discharged from the hospital and was supposed to last 6 months. The majority of patients began AC within 3 months of curative surgery. The treatment regimen included oral S-1 at doses ranging from 80–120 mg per day, depending on body surface area. The medication was taken twice daily for 28 days, followed by a 14-day rest period, and the cycle was repeated every 6 months unless treatment-limiting toxicity occurred.

The relative dose intensity (RDI) was calculated by comparing the actual dose delivered to the intended S-1 dosage [15]. The S-1 therapy was considered complete when oral administration was maintained with an RDI greater than 80% [16]. Hematologic and biochemical parameters, as well as clinical indicators such as body weight changes, were monitored at the start of AC and all subsequent visits.

The postoperative care included monthly laboratory testing and contrast-enhanced computed tomography scans every 3 months at the hospital. AEs were classified using the Common Terminology Criteria for Adverse Events version 5.0, and those with a grade ≥3 were considered severe [17]. In addition, tumor staging was performed based on the 8th edition of the TNM classification system established by the Union for International Cancer Control (UICC), which is widely used for PDAC.

### 2.4. Definition of the GNRI

The GNRI value was calculated from postoperative blood tests performed before the start of AC, using body weight, height, and serum albumin. The GNRI was calculated using the following formula:

GNRI = [14.89 × serum albumin (g/L)] + [41.7 × current/ideal body weight (kg)]. The ideal body weight was calculated as 22 × patient height^2^(m). If the current body weight was greater than the ideal body weight, the ratio was one [18].

The patients were then divided into two groups: S-1 completion and noncompletion, based on a predetermined GNRI threshold value reflecting their nutritional status and potential tolerance to AC.

### 2.5. Statistical Analysis

All statistical analyses were carried out using SPSS^®^ software (version 16.0, SPSS Inc., Chicago, IL, USA) and GraphPad Prism (version 5.0, GraphPad Software Inc., La Jolla, CA, USA). Continuous data with nonparametric distributions were presented as medians with interquartile ranges, whereas categorical data were summarized as frequencies and percentages. As appropriate, to compare groups, we used the χ^2^ test, Fisher’s exact test, or the Mann–Whitney U test.

Univariate analysis was used to determine the independent variables influencing S-1 therapy completion. The cutoff value of the GNRI for predicting the risk of S-1 incompletion was determined based on the cutoff value (GNRI < 94.4) derived from previous studies [13]. Therefore, ROC curve analysis was omitted in the present study. Survival outcomes, specifically recurrence-free survival (RFS), and overall survival (OS) after curative resection were assessed using the Kaplan–Meier method, with differences between the survival curves analyzed using the log-rank test. A Cox proportional hazards model was also used to evaluate the association between the postoperative GNRI and survival outcomes. A *p*-value of < 0.05 was considered statistically significant.

## 3. Results

### 3.1. Patients Enrolled in the Study Undergoing Adjuvant Chemotherapy After Curative Surgery for PDAC

A total of 180 patients with PDAC underwent curative surgery before beginning the S-1 AC regimen during the study period. Among these patients, 93 (51.7%) could continue on AC, resulting in a relative dose intensity (RDI) of more than 80%. In contrast, 48 patients had to reduce their doses or discontinue treatment due to AEs. The remaining 39 patients experienced recurrences during the AC administration period, necessitating a regimen change (Figure 1). Using data from the previous study, the cutoff value for the GNRI was also set at 94.4 in the present study [13].

### 3.2. Comparison of Patients Categorized by GNRIs of < 94.4

Based on the previous study’s cutoff value of 94.4 for GNRIs [13], patients were divided into two groups: The higher GNRI group (≥94.4, *n* = 71) and the lower GNRI group (<94.4, *n* = 70). Table 1 comprehensively summarizes patient characteristics by GNRIs of < 94.4. There were no significant differences between the two groups in sex, age, or carbohydrate antigen 19-9 (CA19-9). However, there was a statistically significant difference between the two groups in terms of body mass index (BMI), surgical procedure, and severe AEs. Furthermore, within these groups, 54 patients (76.1%) in the higher GNRI group and 39 patients (55.7%) in the lower GNRI group completed S-1 AC treatment (*p* = 0.013). Moreover, except for brief interruptions due to missed doses or fever caused by common colds, all patients received S-1 with an RDI of 100% in both groups. In addition, the details of the adverse events are presented in Table 2.

### 3.3. The GNRI and Patient Prognosis

A GNRI value < 94.4 was evaluated for its prognostic impact (Figure 2). Patients with a GNRI of < 94.4 had significantly lower recurrence-free survival (RFS) (HR 1.54; 95% CI 1.04–2.28, *p* = 0.024, MST: 15.6 vs. 26.2) and overall survival (OS) rates (HR 1.89; 95% CI 1.20–2.99, *p* = 0.004, MST: 28.7 vs. 48.9) compared to those with a GNRI of ≥ 94.4.

## 4. Discussion

The current study’s findings validated previous research, demonstrating that a higher postoperative GNRI (≥94.4) is significantly associated with a successful completion of S-1 adjuvant chemotherapy in patients who underwent curative resection for PDAC. In particular, a GNRI of ≥ 94.4 was significantly associated with improved survival outcomes, with a hazard ratio (HR) for recurrence-free survival (RFS) rate of 1.54 (95% confidence interval [CI], 1.04–2.28) and for overall survival (OS) of 1.89 (95% CI, 1.20–2.99). These findings highlight the prognostic utility of the GNRI not only for predicting chemotherapy tolerance but also for forecasting long-term outcomes. As such, the GNRI may serve as a simple yet powerful tool for risk stratification and perioperative nutritional management in clinical practice.

Accumulated evidence highlights the importance of nutritional status in perioperative management. Several studies have found a link between various nutritional markers such as the GNRI, neutrophil-to-eosinophil ratio, neutrophil-to-lymphocyte ratio (NLR), prognostic nutritional index (PNI), and C-reactive protein-to-albumin ratio (CAR) and key clinical outcomes. These indices have been linked to the incidence of POCs, such as pancreatic fistula and surgical site infections, as well as to the prognosis for various cancer types [19,20,21,22,23].

Nutritional status has a significant impact on chemotherapy outcomes, as evidenced by randomized controlled trials that show a clear link between improved nutritional status and increased response rates and reduced AEs during neoadjuvant chemotherapy (NAC) for esophageal and ovarian cancers [24,25]. This relationship extends beyond those cancers, with studies demonstrating its effect on chemotherapy toxicity and efficacy in gastric and rectal cancers [26,27]. However, it is important to note that the relationship between nutritional status and chemotherapy outcomes is still debatable, as conflicting reports have identified potential negative correlations [28,29]. This emphasizes the need for additional research to clarify this complex and multifaceted interaction. Despite a growing body of evidence emphasizing the critical relationship between nutritional status and chemotherapy outcomes, as well as increased attention to this topic in recent years, surgeons’ approaches to nutritional care remain inconsistent. As a result, a significant proportion of patients do not receive appropriate nutritional assessments or necessary nutritional interventions. Furthermore, despite the availability of new evidence, reports indicate that surgeons frequently show a lack of proactive engagement in addressing patients’ nutritional status prior to surgery or chemotherapy [30]. Álvaro Sanz et al. discovered that more than one-third of cancer patients undergoing chemotherapy are candidates for early nutritional intervention [31]. As a result, there is a demand for higher-level evidence demonstrating the use of assessing cancer patients’ nutritional statuses. Furthermore, recent research has shown that nutritional status is important in determining AC efficacy and completion rates [32,33]. Similarly, our institution has reported a link between CAR and the completion rate of S-1 therapy [34,35].

Various nutritional indicators have been used to assess AC tolerance, particularly with S-1, across cancer types. These include markers like the GNRI, PNI, and NLR [13,36,37]. Based on the established links between nutritional status and treatment outcomes, we hypothesized that the GNRI might influence the S-1 AC completion rate in PDAC patients. Our previous study was the first to show a significant correlation between the GNRI and S-1 completion rates [13]. To support these findings, we conducted a validation study with an external cohort, the current study, which confirmed the significance of our preliminary findings.

S-1, an orally administered chemotherapy medication, is widely used in Asian countries, particularly in Japan, to treat various cancers, including PDAC. This medication combines tegafur, a prodrug, with miracil and potassium terbacil, which work together to boost its therapeutic effects. Gimeracil inhibits dihydropyrimidine dehydrogenase, resulting in increased fluorouracil levels in the bloodstream and tumor tissues, whereas potassium terbacil reduces gastrointestinal toxicity by preventing fluorouracil phosphorylation in the gastrointestinal tract [38]. Clinical studies, such as the JASPAC01 trial for PDAC [6] and the biliary tract cancer trial [8], have shown significant survival benefits for Japanese patients receiving S-1 as AC. As a result, S-1 has established itself as a reliable and effective treatment option for postoperative AC in PDAC patients, particularly in Japan and other Asian countries.

Clinical studies have shown that a significant number of postoperative PDAC patients have difficulty completing S-1 AC therapy due to AEs or POCs. The previous cohort study reported an S-1 completion rate of 69.5% [13], and the current validation study found a similar 70.5% completion rate. With nearly 30% of patients unable to complete the treatment, identifying a reliable, noninvasive marker to predict S-1 noncompletion due to AEs is critical. Such a marker would not only provide valuable insights into the patient’s prognosis but would also help determine appropriate postoperative monitoring intervals.

The current study’s findings reaffirmed the link between a GNRI value below the established threshold and an increased risk of S-1 therapy noncompletion due to AEs. These findings highlight the importance of optimizing nutritional status before initiating AC to potentially improve treatment adherence. Furthermore, both studies consistently identified the GNRI as a predictor of S-1 therapy completion, indicating that it may be useful as a prognostic indicator. Notably, the GNRI thresholds were found to approximate the values identified in previous studies conducted at our institution, highlighting their predictive utility for POCs in PDAC patients [39]. Moreover, a review of the literature on various cancers, including pancreatic, gastric, and lung cancers, revealed that GNRI cutoff values generally range from 92 to 98, further supporting the appropriateness of the threshold used in the present study [40,41,42,43,44]. This agreement emphasizes the dual importance of the GNRI in assessing the risks of both postoperative complications and AC noncompletion due to AEs. Numerous previous studies have found that nutritional indices, including the GNRI, have strong associations with prognosis, POC risk, and chemotherapy-related AEs in several cancers, as evidenced by systematic reviews and meta-analyses [45,46,47,48,49]. However, conflicting evidence from some studies suggests that there is no direct link between nutritional status and prognosis. This emphasizes the importance of validation through large-scale, multi-institutional studies to confirm these findings and establish the GNRI’s definitive clinical role [50,51].

Although this study validates previous findings, there are several limitations to consider. First, the single-center design and small sample size may have introduced biases, limiting the findings’ generalizability and the GNRI’s reliability as a predictive marker. Second, because the study was conducted retrospectively, there is a possibility of selection bias, as patient inclusion was based on historical records rather than prospective enrollment. Third, variations in the S-1 treatment protocols—such as the timing of initiation, dosing, dose adjustments, and discontinuation—were left to the discretion of the attending surgeons, which may have influenced the consistency of treatment outcomes. To address these limitations, additional validation through multi-center studies with larger, more diverse cohorts is required to confirm the GNRI’s utility in predicting S-1 therapy completion and clinical outcomes.

## 5. Conclusions

Consistent with our previous findings, this study found that a lower postoperative GNRI is a significant predictor of noncompletion of S-1 AC therapy due to AEs. In line with previous findings, the current analysis confirmed that GNRI values below 94.4 are strongly correlated with poorer clinical outcomes, highlighting its role as a reliable prognostic marker in this setting.

## Figures and Tables

**Figure 1 cancers-17-01448-f001:**
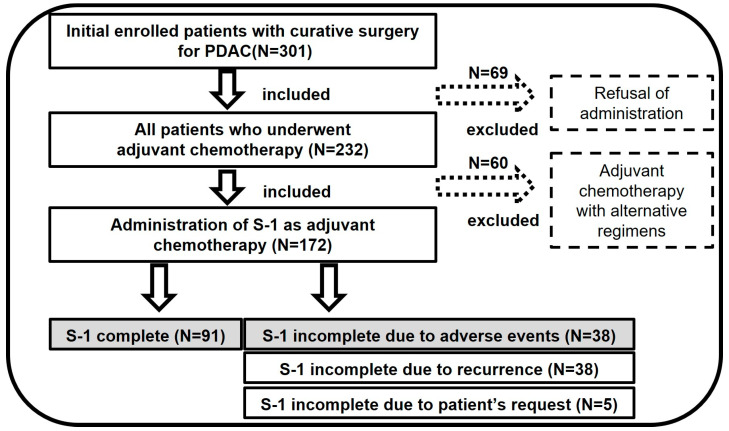
Patient selection diagram. This retrospective study included 309 patients with pancreatic ductal adenocarcinoma (PDAC) who underwent curative surgery between 2010 and 2023. A total of 129 patients were excluded; 69 declined S-1 and 60 received other therapies. The final analysis was conducted on 180 patients.

**Figure 2 cancers-17-01448-f002:**
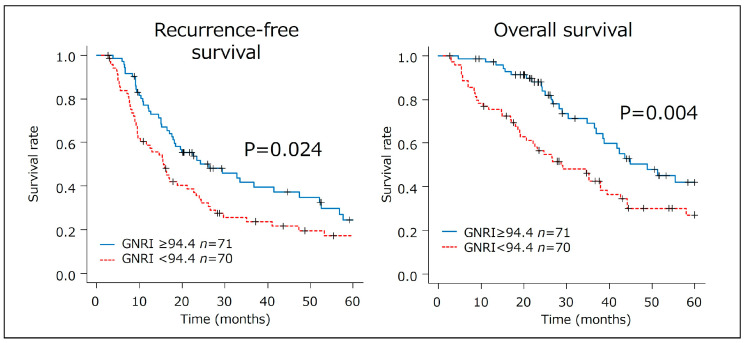
Role of GNRI of ≥ 94.4 in predicting recurrence-free and overall survival in patients with PDAC treated with S-1 AC therapy.

**Table 1 cancers-17-01448-t001:** Differences in perioperative factors based on the GNRI cutoff value.

Characteristics	GNRI ≥ 94.4 (*n* = 71)	GNRI < 94.4 (*n* = 70)	*p* Value
Sex (male)	42 (59.2%)	35 (50.0%)	0.312
Age (year)	68 (43–84)	68 (45–82)	0.851
Body mass index (kg/m^2^)	24.3 (18.6–36.8)	20.8 (14.0–31.8)	<0.001
NAC	40 (56.3%)	37 (52.9%)	0.736
Operation type			<0.001
DP	34 (47.9%)	9 (12.9%)	
PD	35 (49.3%)	56 (80.0%)	
TP	2 (2.8%)	5 (7.1%)	
Pathological stage			0.671
1	8 (15.8%)	7 (10.0%)	
2	62 (87.3%)	60 (85.7%)	
3	1 (1.4%)	3 (4.3%)	
Severe POCs	18 (25.4%)	19 (27.1%)	0.565
Alb, mg/dL	3.8 (3.2–4.7)	3.0 (1.9–3.8)	<0.001
CA19-9 (U/mL)	20.9 (1.2–12,000)	67 (1.2–6450)	0.867
AC complete	54 (76.1%)	39 (55.7%)	0.013
Severe AEs	9 (12.7%)	19 (27.1%)	0.036

NAC: neoadjuvant chemotherapy, DP: distal pancreatectomy, PD: pancreatoduodenectomy, TP: total pancreatectomy, POC: postoperative complication, CA: carbohydrate antigen, AC: adjuvant chemotherapy, AE: adverse event.

**Table 2 cancers-17-01448-t002:** Details of severe adverse events in groups stratified by the GNRI cutoff value.

	GNRI ≥ 94.4	GNRI < 94.4
Severe Adverse Events	*N* = 9	*N* = 19
Stomatitis	0	4
Liver dysfunction	0	0
Diarrhea	1	2
Lacrimal duct stenosis	0	0
Dermatitis	2	0
General fatigue	2	4
Thrombocytopenia	1	1
Anorexia	3	4
Sepsis	1	1
Neutropenia	1	1
Interstitial pneumonia	0	1

There are cases where symptoms overlap.

## Data Availability

The datasets used and analyzed during the current study are available from the corresponding author upon reasonable request.

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
