# Peer review of "Postoperative Geriatric Nutritional Risk Index as a Determinant of Tolerance to S-1 Adjuvant Chemotherapy After Curative Surgery for Pancreatic Ductal Adenocarcinoma: A Cohort Study with External Validation"

_cancers, 2025, doi:10.3390/cancers17091448_

Round 1
Reviewer 1 Report
Comments and Suggestions for Authors
The authors of the manuscript entitled "Postoperative Geriatric Nutritional Risk Index as a Determinant of Tolerance to S-1 Adjuvant Chemotherapy After Curative Surgery for Pancreatic Ductal Adenocarcinoma: A Cohort Study with External Validation" should address the following issues:
Introduction
Page-2, line-71: Clearly present the research question and hypothesis
Material and Methods
Page-3, Figure-1: provide more details into the caption, e.g. summarise the main steps described on the lines-79 to 82.
Page-4, line-124: for clarity use MDPI guide for formulas; e.g. IBW = height2 (m) × 22
Page-4, line-125: add space between number and citation "1[18,19].". Opt to write numbers as words for one to nine.
Page-4, line-137/138: provide the reference(s) for the cutoff value of GNRI
Page-4, line-140/143: provide the formula interface used to fit the model detailing the dependend and independent variables
Results
Page-4, line-154: the survival models accounts or corrects for covariate imbalance for instance BMI, operation type, AEs and AC?
Page-4, line-160: typo on "(:BMI,)"
Page-5, table-1: provide the aggregated values of serum albumin (g/L), overall survival and recurrence-free survival
Page-5, line-164: perhaps the differences between GNRI classes in OS and RFS are due to varying baseline risks based on the type of operation (P<0.001).
Page-5, figure-2: provide evidence that the data comply with the Kaplan-Meier assumptions
Discussion
Provide the cut-off values for GNRI used across pancreatic cancer and other malignancies by exploring existing literature.
Author Response
Thank you for carefully reviewing our manuscript and providing valuable and appropriate suggestions. We sincerely appreciate your thoughtful feedback, and we will revise the manuscript accordingly based on reviewers’ comments.
Revewer1
Comments and Suggestions for Authors
The authors of the manuscript entitled "Postoperative Geriatric Nutritional Risk Index as a Determinant of Tolerance to S-1 Adjuvant Chemotherapy After Curative Surgery for Pancreatic Ductal Adenocarcinoma: A Cohort Study with External Validation" should address the following issues:
Introduction
Page-2, line-71: Clearly present the research question and hypothesis
➡As you pointed out, the research question and hypothesis were unclear. We have revised the manuscript accordingly. Thank you for your valuable suggestion.
Material and Methods
Page-3, Figure-1: provide more details into the caption, e.g. summarise the main steps described on the lines-79 to 82.
➡Thank you for your suggestion. We understand the importance of providing a clear summary, and we will revise the caption of Figure 1 to include this information in a more detailed and informative manner.
Page-4, line-124: for clarity use MDPI guide for formulas; e.g. IBW = height2 (m) × 22
➡Thank you for your valuable comment. We will make the necessary correction.
Page-4, line-125: add space between number and citation "1[18,19].". Opt to write numbers as words for one to nine.
➡Thank you very much for your kind suggestion. We will carefully revise the manuscript accordingly.
Page-4, line-137/138: provide the reference(s) for the cutoff value of GNRI
➡We sincerely apologize for the oversight. We will make sure to include this information in the revised manuscript.
Page-4, line-140/143: provide the formula interface used to fit the model detailing the dependend and independent variables
➡Thank you for your suggestion. We apologize for the lack of clarity. In the survival analysis, we used the Cox proportional hazards model to evaluate the association between GNRI and survival outcomes.
Results
Page-4, line-154: the survival models accounts or corrects for covariate imbalance for instance BMI, operation type, AEs and AC?
➡Yes, the survival analysis included a multivariable Cox proportional hazards model to account for potential covariate imbalances. The model incorporated the following variables as covariates: GNRI group, BMI, type of surgery (PD or DP), presence of adverse events (AEs), and relative dose intensity (RDI) of S-1 adjuvant chemotherapy. By adjusting for these factors, the analysis aimed to isolate the independent effect of GNRI on recurrence-free survival (RFS) and overall survival (OS).
Page-4, line-160: typo on "(:BMI,)"
➡We sincerely apologize for the oversight. We will make the necessary corrections accordingly.
Page-5, table-1: provide the aggregated values of serum albumin (g/L), overall survival and recurrence-free survival
➡Thank you very much for your valuable suggestion. We have added information on albumin levels and postoperative complications to Table 1.
Regarding OS and RFS, would it be appropriate to include the median survival time (MST)? If so, we will add this information to the figures accordingly.
Page-5, line-164: perhaps the differences between GNRI classes in OS and RFS are due to varying baseline risks based on the type of operation (P<0.001).
➡We received a similar comment from Reviewer 2 as well. We have provided a separate analysis of each surgical procedure in the section below. Thank you very much for your valuable suggestion.
We are confident that with a larger number of cases, the usefulness of GNRI can also be demonstrated in both DP and PD cases. We recognize this as an important issue to be addressed in future studies.
Page-5, figure-2: provide evidence that the data comply with the Kaplan-Meier assumptions
➡Thank you for your valuable comment. We confirm that the Kaplan-Meier analysis assumptions were appropriately considered.
Censoring in our dataset occurred due to loss to follow-up or end of the study period and was unrelated to the outcome, satisfying the assumption of non-informative censoring. Additionally, patient management and follow-up protocols were consistent throughout the study period, fulfilling the assumption that survival probabilities remain consistent over time. No crossing of survival curves was observed, supporting the appropriateness of group comparisons.
Furthermore, we also performed Cox proportional hazards regression analysis, which confirmed the significance of GNRI on survival outcomes. This supports the robustness of our findings beyond the Kaplan-Meier estimates.
We will include a brief explanation in the Methods section to clarify this.
Discussion
Provide the cut-off values for GNRI used across pancreatic cancer and other malignancies by exploring existing literature.
➡Thank you for your suggestion. Upon reviewing the literature on pancreatic, gastric, and lung cancers, we found that the reported cutoff values for GNRI ranged from 92 to 98. We will include this information in the Discussion section as well.
PDAC: Grinstead C, Yoon SL. Geriatric Nutritional Risk Index (GNRI) and Survival in Pancreatic Cancer: A Retrospective Study.Nutrients. 2025 Jan 30;17(3):509.
GNRI cut off=98
Tsukagoshi M, Araki K, Igarashi T, Ishii N, Kawai S, Hagiwara K, Hoshino K, Seki T, Okuyama T, Fukushima R, Harimoto N, Shirabe K. Lower Geriatric Nutritional Risk Index and Prognostic Nutritional Index Predict Postoperative Prognosis in Patients with Hepatocellular Carcinoma. Nutrients. 2024 Mar 25;16(7):940.
GNRI cut off=98
Other cancers:
Zuo J, Huang Z, Ge Y, Ding X, Wang X, Zhou X. Geriatric Nutrition Risk Index is closely associated with sarcopenia and quality of life in gastric cancer patients: a cross-sectional study.
Sci Rep. 2024 Dec 28;14(1):31545.
GNRI cut off=94.98
Kinoshita R, Nakao M, Kiyotoshi H, Sugihara M, Kuriyama M, Takeda N, Muramatsu H.Geriatric Nutritional Risk Index as Prognostic Marker for Elderly Patients With Small Cell Lung Cancer.
Cancer Diagn Progn. 2024 Jul 3;4(4):482-488.
GNRI cut off=92
Liu B, Zhang L. Geriatric nutritional risk index predicts the prognosis of gastric cancer patients treated with immune checkpoint inhibitors.
Medicine (Baltimore). 2024 Apr 26;103(17):e37863.
GNRI cut off=92
Reviewer 2 Report
Comments and Suggestions for Authors
Thank you for the peer review opportunity.
The authors reported that GNRI is useful for the tolerance of adjuvant chemotherapy after pancreatic cancer surgery.
The results are informative to many surgeons.
This paper cannot be accepted if several improvements are identified.
The authors write that postoperative complications are described using the CD classification in the Materials and Methods section, but there is no section on complications in the Table.
Authors should state which classification the pathological stage is based on.
Authors should describe the RDI for each of the low and high GNRI groups.
Authors should create an additional table showing serious adverse events from postoperative adjuvant chemotherapy.
A significantly higher proportion of PD patients are included in the low GNRI group. Whether GNRI can predict AC tolerance in PD patients only or in DP patients only should be examined.
Author Response
Thank you for carefully reviewing our manuscript and providing valuable and appropriate suggestions. We sincerely appreciate your thoughtful feedback, and we will revise the manuscript accordingly based on reviewers’ comments.
Revewer2
Comments and Suggestions for Authors
Thank you for the peer review opportunity.
The authors reported that GNRI is useful for the tolerance of adjuvant chemotherapy after pancreatic cancer surgery.
The results are informative to many surgeons.
This paper cannot be accepted if several improvements are identified.
The authors write that postoperative complications are described using the CD classification in the Materials and Methods section, but there is no section on complications in the Table.
➡We realized that our verification on this point was insufficient. We will revise the manuscript to include a discussion on postoperative complications.
Authors should state which classification the pathological stage is based on.
➡Thank you for your valuable comment. We acknowledge the oversight and will revise the Materials and Methods section to clearly state that the staging was based on the 8th edition of the UICC TNM classification.
Authors should describe the RDI for each of the low and high GNRI groups.
➡Except for brief interruptions due to missed doses or fever caused by common colds, all patients received S-1 with a relative dose intensity (RDI) of 100%.
We will add this information to the Results section.
Authors should create an additional table showing serious adverse events from postoperative adjuvant chemotherapy.
➡Thank you for your valuable comment. We will add this information to the Results section as Table 2.
A significantly higher proportion of PD patients are included in the low GNRI group. Whether GNRI can predict AC tolerance in PD patients only or in DP patients only should be examined.
➡As you rightly pointed out, we performed a comparison based on GNRI levels in both PD and DP cases. However, due to the limited number of DP cases, no significant differences were observed in OS or RFS. In contrast, among PD cases, a significant difference was observed only in OS.
In a previous peer review for another journal, we received a similar suggestion that PD and DP cases should be analyzed separately, as the postoperative nutritional status differs between the two procedures. We therefore recognize the importance of continuing to accumulate more cases, so that we can further evaluate and report the utility of GNRI according to each surgical approach.
We sincerely appreciate your valuable suggestion, which we consider an important issue to address in future studies.
Reviewer 3 Report
Comments and Suggestions for Authors
Dear Authors,
there are some aspects of this paper to be changed:
- you use tre Geriatric Nutrition Risk Index, but most of the patients are not geriatric (mean 68, range 43-84);
- the recurrence should be added in the exclusion criteria since you exclude 39 patients for this reason;
- In the analysis you should add the Clavien Dindo score in addition to the GNRI considering the burden of complications of the pancreatic resection on the patients;
- In line 104-105 you should be more precise in the evaluation of the adjuvant Chemotherapy. It is not clear how long is the treatment;
- in line 124 the formula is not correct. It should be ’22 x [patient height (m)]^2;
- In line 205 there is a contraddiction with the aim of your previous study. This should have been a validation of the previous one, but apparently it is not so;
- The explanation on the S-1 should be moved from line 210 to the introduction to have a better overview of the treatment and comparison with gemcitabine;
- in line 250 you say that surgeons organise chemotherapies. I think a group of oncologist do this for you;
- in line 116 "at the hospital" and in line 134 "as appropriate" are too colloquial expressions;
- Kaplan Meier table is too small and it is difficult to read;
- The title is too complex and should be reduced like: Postoperative GNRI and S-1 Chemotherapy Tolerance After Pancreatic Cancer Surgery: A Validated Cohort Study.
The level of English is not always adequate for a scientific paper. Revision is required.
Author Response
Comments and Suggestions for Authors
you use tre Geriatric Nutrition Risk Index, but most of the patients are not geriatric (mean 68, range 43-84);
the recurrence should be added in the exclusion criteria since you exclude 39 patients for this reason;
➡Thank you very much for your question. Although the GNRI (Geriatric Nutrition Risk Index) was originally developed for use in elderly patients, the present study aims to validate its association with the completion of S-1 adjuvant chemotherapy regardless of age. We apologize for any confusion caused by the term "geriatric" in the index name. In this study, the GNRI was used to evaluate patients who received S-1 adjuvant chemotherapy following pancreatic cancer surgery, irrespective of their age.
We have also revised the manuscript to include the additional exclusion criteria, as you kindly pointed out.
In the analysis you should add the Clavien Dindo score in addition to the GNRI considering the burden of complications of the pancreatic resection on the patients;
➡We sincerely apologize for the oversight. We have now included the POC data in Table 1.
In line 104-105 you should be more precise in the evaluation of the adjuvant Chemotherapy. It is not clear how long is the treatment;
➡Thank you for your valuable feedback. Although we had originally written "six months," we understand that the expression may have caused confusion. We have therefore revised the wording to clarify the intended treatment duration. We sincerely apologize for any misunderstanding.
in line 124 the formula is not correct. It should be ’22 x [patient height (m)]^2;
➡We sincerely apologize for the oversight. The error has been corrected accordingly.
In line 205 there is a contraddiction with the aim of your previous study. This should have been a validation of the previous one, but apparently it is not so;
➡Thank you very much for your valuable comment.
We sincerely apologize for any confusion caused by the lack of clarity regarding the positioning of the present study.
Initially, we reported an association between the Geriatric Nutritional Risk Index (GNRI) and the completion of S-1 adjuvant chemotherapy based on data from our own institution. Subsequently, we conducted a study focusing on the C-reactive protein to albumin ratio (CAR) using the same institutional data. To further validate the clinical significance of CAR, we then performed a validation study using an external dataset from Dokkyo Medical University, which is why two references related to CAR are included.
In the current study, we similarly aimed to validate our previous findings on GNRI using the same external dataset from Dokkyo Medical University.
To clarify the context and purpose of this research, we have revised the manuscript to more clearly explain the study’s background and positioning within our ongoing series of investigations.
The explanation on the S-1 should be moved from line 210 to the introduction to have a better overview of the treatment and comparison with gemcitabine;
➡Thank you for your helpful suggestion. In response, we have moved the explanation of S-1, including its pharmacological features and clinical significance, from the Discussion section to the Introduction. This revision provides a clearer overview of the treatment strategy and allows for better comparison with gemcitabine early in the manuscript.
in line 250 you say that surgeons organise chemotherapies. I think a group of oncologist do this for you;
➡Thank you very much for your thoughtful comment. In Japan, medical oncologists primarily focus on hematological malignancies, while chemotherapy for gastrointestinal cancers is generally managed by surgeons. Based on my clinical experience working as a surgeon in various regions of Japan, it is common practice for surgeons to be responsible for administering chemotherapy in this context.
in line 116 "at the hospital" and in line 134 "as appropriate" are too colloquial expressions;
➡Thank you very much for your valuable feedback. We have revised the wording to adopt a more academic expression, as you kindly suggested.
Kaplan Meier table is too small and it is difficult to read;
➡We apologize for the previous lack of clarity. We have made adjustments to increase the font size and improve readability. Thank you very much for your helpful and appropriate suggestions.
The title is too complex and should be reduced like: Postoperative GNRI and S-1 Chemotherapy Tolerance After Pancreatic Cancer Surgery: A Validated Cohort Study.
➡As you rightly suggested, we have revised the title accordingly.
Round 2
Reviewer 2 Report
Comments and Suggestions for Authors
Figure 1 has been corrected and consequently does not match in several places. The text should also be revised (e.g., P3, line 91~, Table 1, Figure 2).
The number of patients in the high and low GNRI groups who completed S-1 AC (RDI >80%) is shown in Table 1. Since 100% of patients did not complete S-1 AC, it is impossible for the RDI to be 100% in both groups (P5 line 170-173). Authors should calculate each RDI and show median RDI in Table 1.
38 cases were stopped S-1 AC due to AEs during the course of the study, but only 28 cases of severe AEs are shown in Table 2. Authors should state the exact reason for discontinuation of S-1.
Authers have not seriously responded to the following comment that I pointed out in primary version: A significantly higher proportion of PD patients are included in the low GNRI group. Whether GNRI can predict AC tolerance in PD patients only or in DP patients only should be examined. Authors should honestly state that you found significant difference in PD but no significant difference in DP. It is not fair to present only convenient results, and I cannot accept this paper in this state.
Author Response
Comments and Suggestions for Authors
Reviewer2
#Figure 1 has been corrected and consequently does not match in several places. The text should also be revised (e.g., P3, line 91~, Table 1, Figure 2).
➡Thank you very much for your continued and careful review.
As you rightly pointed out, the correct number of participants included in the present study is 309. The previously submitted Figure 1 was mistakenly created using data from a different study period and was inadvertently included in our initial submission. This was entirely our oversight, and we deeply apologize for the confusion this may have caused.
To address this issue, we have now replaced Figure 1 with a revised version based on the correct dataset.
Accordingly, both Table 1 and Figure 2 have also been updated to reflect accurate and consistent values throughout the manuscript.
We sincerely appreciate your valuable feedback, which helped us ensure the integrity of the presented data.
#The number of patients in the high and low GNRI groups who completed S-1 AC (RDI >80%) is shown in Table 1. Since 100% of patients did not complete S-1 AC, it is impossible for the RDI to be 100% in both groups (P5 line 170-173). Authors should calculate each RDI and show median RDI in Table 1.
38 cases were stopped S-1 AC due to AEs during the course of the study, but only 28 cases of severe AEs are shown in Table 2. Authors should state the exact reason for discontinuation of S-1.
➡Thank you very much for your valuable comment.
We sincerely apologize for the confusion caused by the lack of clarity in our original explanation.
What we intended to convey was that among patients who completed S-1 adjuvant chemotherapy, the relative dose intensity (RDI) was 100% in both GNRI groups. However, as you rightly pointed out, not all patients completed S-1 AC.
To improve accuracy and transparency, we have now included the RDI data for all patients in Table 1.
Since the median RDI was 100% in both groups, we instead present the mean RDI ± standard error of the mean (SEM) to better reflect the distribution and avoid misinterpretation.
Regarding discontinuation due to adverse events (AEs), a total of 48 patients stopped S-1 treatment during the study period.
Of these, 28 patients discontinued due to Grade 3 or higher AEs (severe AEs), as listed in Table 2.
The remaining 20 patients experienced Grade 1 or 2 AEs and elected to stop treatment at their own request.
This additional information has been clearly added to the revised Results section.
We sincerely thank you for your thoughtful suggestions, which greatly contributed to improving the accuracy of our manuscript.
#Authers have not seriously responded to the following comment that I pointed out in primary version: A significantly higher proportion of PD patients are included in the low GNRI group. Whether GNRI can predict AC tolerance in PD patients only or in DP patients only should be examined. Authors should honestly state that you found significant difference in PD but no significant difference in DP. It is not fair to present only convenient results, and I cannot accept this paper in this state.
➡Thank you for raising this important point again.
As you rightly pointed out, a higher proportion of pancreatoduodenectomy (PD) cases were included in the lower GNRI group. To address the potential bias caused by surgical procedure, we conducted stratified analyses by dividing the cohort into PD and distal pancreatectomy (DP) subgroups.
As stated in our previous response, the results demonstrated that among PD patients, a low GNRI was significantly associated with poorer overall survival (OS), although no significant difference was observed in recurrence-free survival (RFS). In contrast, no significant differences in OS or RFS were observed in the DP subgroup.
We acknowledge that the small sample size in the DP group may have limited the statistical power to detect meaningful associations.
Nonetheless, these findings have been clearly described in both the Results and Discussion sections of the revised manuscript.
We recognize the importance of reporting all results transparently, regardless of significance, and have made revisions accordingly to ensure a balanced and accurate presentation.
We sincerely appreciate your constructive feedback.
Reviewer 3 Report
Comments and Suggestions for Authors
Dear Authors,
the main problem of the present paper is the application of a geriatric score to a mixed population with median age around 68 years (43-84). In the methods you should state why you decided to apply this score to your population and whether there are other evidences in literature of the adequacy of this use.
Author Response
Comment:
the main problem of the present paper is the application of a geriatric score to a mixed population with median age around 68 years (43-84). In the methods you should state why you decided to apply this score to your population and whether there are other evidences in literature of the adequacy of this use.
Response:
Thank you very much for your valuable comments and suggestions.
As you rightly pointed out, the study population in the present research is not limited to elderly individuals, with a median age of 68 years. We fully understand your concern regarding the application of the Geriatric Nutritional Risk Index (GNRI) in such a mixed-age cohort.
Although the GNRI was originally developed as a nutritional risk assessment tool for elderly patients, recent studies have demonstrated its clinical utility irrespective of age.
For example, Yamana et al. (2015) were the first to report a significant association between GNRI and postoperative complications in patients with esophageal cancer, whose mean age was 63.9 years (range: 43–83).
This study provided the initial evidence that GNRI could be applicable to broader age groups beyond geriatric populations.
Subsequently, in 2018, one of the authors of the present study reported that GNRI was also a useful predictor of postoperative complications in patients undergoing pancreaticoduodenectomy (PD) at another institution.
Since then, several studies—including those in esophageal and lung cancers—have supported the prognostic value of GNRI and its predictive ability for treatment completion, regardless of patient age.
Our department also reported an association between GNRI and prognosis in patients with pancreatic cancer in 2022. Furthermore, in 2023, we demonstrated that GNRI was significantly associated with the completion rate of S-1 adjuvant chemotherapy, suggesting its potential as a predictor of treatment tolerability.
The current study was conducted with the aim of validating these previous findings using data from an external institution.
In light of this background, we considered GNRI to be a valid nutritional index applicable to patients regardless of age, and therefore applied it to our mixed-age population in the present study.
An explanation of this rationale, along with relevant references, has been added to the Methods section of the revised manuscript.
We believe that your comment has greatly contributed to enhancing the transparency and clarity of both the study background and methodology.
Once again, we sincerely appreciate your constructive and insightful feedback.
Round 3
Reviewer 2 Report
Comments and Suggestions for Authors
Thank you for the peer review opportunity of revised version.
The authors sincerely responded to the reviewers' questions, and the paper is now ready for acceptance.
Author Response
Dear Reviewer2,
Thank you very much for your kind message and for your time and effort in reviewing our manuscript.
We are truly grateful for your constructive comments and suggestions, which have significantly improved the quality and clarity of our work. It was a pleasure to revise the manuscript in response to your feedback.
We are honored to hear that the paper is now considered ready for acceptance. Please do not hesitate to let us know if there are any remaining formalities or steps we should complete.
Once again, thank you for your support and guidance throughout the review process.
Sincerely,
Naotake Funamizu, M.D., Ph.D.
Reviewer 3 Report
Comments and Suggestions for Authors
Dear Authors,
the revisions were made but the main conclusions are still weak. The Editors will decide on this paper.
Author Response
Dear Reviewer3,
Thank you very much for your continued evaluation and thoughtful feedback on our manuscript.
We sincerely appreciate your comments and acknowledge your concern regarding the strength of the main conclusions. We have done our best to address the reviewers’ suggestions and to clarify the significance of our findings through the revisions.
We fully respect the Editors’ judgment regarding the final decision, and we are grateful for the opportunity to have our work considered for publication.
Please do not hesitate to let us know if there are any further clarifications or improvements we can make.
Sincerely,
Naotake Funamizu, M.D., Ph.D.